# SuperFed: Weight Shared Federated Learning

## Abstract

Federated Learning (FL) is a well-established technique for privacy preserving distributed training. Much attention has been given to various aspects of FL training. A growing number of applications that consume FL-trained models, however, increasingly operate under dynamically and unpredictably variable conditions, rendering a single model insufficient. We argue for training a global "family of models" cost efficiently in a federated fashion. Training them independently for different tradeoff points incurs $\approx O(k)$ cost for any $k$ architectures of interest, however. Straightforward applications of FL techniques to recent weight-shared training approaches is either infeasible or prohibitively expensive. We propose SuperFed— an architectural framework that incurs $O(1)$ cost to co-train a large family of models in a federated fashion by leveraging weight-shared learning. We achieve an order of magnitude cost savings on both communication and computation by proposing two novel training mechanisms: (a) distribution of weight-shared models to federated clients, (b) central aggregation of arbitrarily overlapping weight-shared model parameters. The combination of these mechanisms is shown to reach an order of magnitude (9.43x) reduction in computation and communication cost for training a $5 * 10^8$-sized family of models, compared to independently training as few as $k = 9$ DNNs without any accuracy loss.

## 1 Introduction

With the increase in the computational power of smartphones, the use of on-device inference in mobile applications is on the rise, ranging from image recognition (google vision; azure vision) , virtual assistant (Alexa) , voice recognition (google ASR) to recommendation systems (Bin et al., 2019). Indeed, on-device inference is pervasive, especially with recent advances in software (Chen et al., 2018; torch mobile), accelerators (samsung exynos; apple neural engine), and neural architecture optimizations (Howard et al., 2019; Sun et al., 2020; Wu et al., 2019a). The surge in its use cases (Cai et al., 2017; Han et al., 2019; Kang et al., 2017; Lane et al., 2016; Reddi et al., 2021; Wu et al., 2019b) has led to a growing interest in providing support not only for on-device inference, but also for on-device training of these models (Dhar et al., 2021).

Federated Learning (FL) is an emerging distributed training technique that allows smartphones with different data sources to collaboratively train an ML model (McMahan et al., 2017; Chen & Chao, 2020; Wang et al., 2020; Karimireddy et al., 2021; Konečný et al., 2016). FL enjoys three key properties, it — a) has smaller communication cost, b) is massively parallel, and c) involves no data-sharing. As a result, numerous applications such as GBoard (Hard et al., 2018), Apple's Siri (sir, 2019), pharmaceutical discovery (CORDIS., 2019), medical imaging (Silva et al., 2019), health record mining (Huang & Liu, 2019), and recommendation systems (Ammad-ud-din et al., 2019) are readily adopting federated learning.

However, adoption of FL in smartphone applications is non-trivial. As a result, recent works pay attention to the emerging challenges that occur in training, such as data heterogeneity (Karimireddy et al., 2021; Li et al., 2020; Acar et al., 2021), heterogeneous resources (Alistarh et al., 2017; Ivkin et al., 2019; Li et al., 2020; Konečný et al., 2016), and privacy (Truex et al., 2019; Mo et al., 2021; Gong et al., 2021). These helped FL adoption, particularly in challenging training conditions. However, the success of FL adoption depends not only on tackling challenges that occur in training but also *post-training* (deployment). Indeed, deploying ML models for on-device inference is exceedingly challenging (Wu et al., 2019b; Reddi et al., 2021). Yet, most of the existing training techniques in FL do not take these deployment challenges into consideration. In this paper, we focus on developing FL

training algorithms specifically designed to address deployment challenges related to post-training inference.

It is well-established that any single model statically chosen for on-device inference is sub-optimal. This is because the deployment conditions may continuously change on a multi-task system like smartphones (Xu et al., 2019) due to dynamic resource availability (Fang et al., 2018). For instance, the computational budget may vary due to excessive consumption by background apps; the energy budget may vary if the smartphone is in low power or power-saver mode (Yu et al., 2019). Furthermore, an increasing number of applications require flexibility with respect to resource-accuracy trade-offs in order to efficiently utilize the dynamic resources in deployment (Fang et al., 2018). In all of these deployment scenarios, a single model neither satisfies variable constraints nor offers the flexibility to make trade-offs.

In contrast to existing FL approaches that produce a single model, we need to produce multiple model variants (varying in size/latency) for efficient on-device inference. However, training these model variants independently is computationally prohibitive (Cai et al., 2020). This is particularly true for FL, where training these variants independently will cumulatively inflate the communication cost as well as the computation cost. Thus, it is imperative to develop techniques for *training multiple models in a federated fashion cost efficiently without any accuracy loss—achieving asymptotic cost improvements relative to independently training them.*

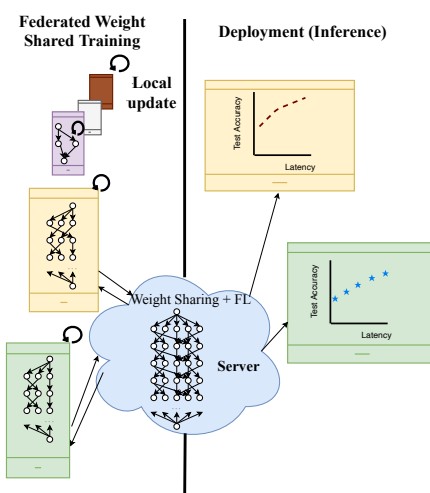

To achieve this goal, we propose SuperFed- a novel federated framework that targets the problem of efficient on-device inference on smartphones with better training algorithms. SuperFed co-trains a family of model variants in a federated fashion *simultaneously* by leveraging weight-sharing (Cai et al., 2020; Yu et al., 2020). After federated training, the clients perform local neural architecture search to find the appropriate model variants for their deployment scenarios. In weight-sharing, all model variants are sub-networks of a supernetwork (Cai et al., 2020) and share their parameters partially. The largest subnetwork's (or supernetwork's) parameters contain other subnetworks' parameters within it as proper subgraphs. There are two key benefits that weight-sharing brings in FL, it a) significantly reduces the communication and computation cost for training k model variants, and b) requires no re-training after the federated training of the supernetwork is complete. Hence, SuperFed decouples training from neural architecture search which allows local clients to dynamically select subnetworks of their choice from the globally-trained supernetwork without any re-training.

However, applying existing weight-shared training techniques to federated learning is challenging. First, weight-shared training techniques like Progressive Shrinking (PS) (Cai et al., 2020) work on centralized i.i.d data, whereas the

Figure 1: **Weight shared FL Training**. Shared weights reside on the server. NN subnetworks are distributed (left) and deployed (right) to participating clients, globally training a dense accuracy-latency trade-off space.

data is decentralized and typically non-i.i.d in FL. Second, PS uses a pre-trained largest subnetwork during the weight-shared training. This requirement becomes impractical in the context of FL as it - a) is hard to obtain a globally pre-trained FL model, or b) may significantly increase the overall communication cost to train it first. Third, weight-sharing training techniques need to minimize interference (Cai et al., 2020; Xu et al., 2022). Interference occurs when smaller subnetworks interfere with the training of larger subnetworks (Fig 2a in Yu et al. (2020)). To mitigate interference, PS adopts multi-phase training approach that prioritizes the training of larger subnetworks before training smaller subnetworks. Such multi-phase training may lead to significant communication cost in federated learning. Instead, we argue that the weight-shared training technique in FL must be be one-shot (single phase) to mitigate interference.

As a part of SuperFed framework, we propose MaxNet — a weight-shared training technique for FL. Figure 1 provides a high level overview of our proposed approach. MaxNet hosts the supernetwork in the server and assumes no prior pre-trained model before the federated training. MaxNet decides *which* individual subnetworks are distributed for training on participating clients and *when* (subnetwork distribution). MaxNet 's subnetwork distribution optimizes both lower bound (smallest subnet)

and upper bound (largest subnet) to increase the accuracy of every subnet in between. Since the subnetworks partially share their parameters, MaxNet also introduces a novel shared model weight aggregation mechanism that mitigates interference in weight-shared FL training. To summarize, our contributions are as follows:

- SuperFed: A weight shared training framework for FL that trains a family of model variants (DNN models), cost-efficiently in a federated fashion.
- MaxNet's subnetwork distribution: a heuristic that trains the upper and lower bounds in the model family by optimizing both bounds and load balancing their distribution over time.
- MaxNet's Aggregation: A mechanism for subnetwork weight aggregation with variable overlaps mitigating intra-network interference.

We perform rigorous evaluation of SuperFed 's weight-shared training technique MaxNet against non-weight shared baselines, where each model variant is trained independently in a federated fashion over CIFAR10/100 and CINIC-10 (downsampled ImageNet images) datasets. MaxNet trains a family of $\approx 5 * 10^8$ subnets showing 9.43x lower computational and 10.94x lower communication cost compared to training as few as 9 subnetworks separately with FedAvg. This order of magnitude reduction in training cost is achieved with no accuracy loss compared to independently trained subnetworks (without weight-sharing) with FedAvg.

## 1.1 Related Work

In this section, we describe FL approaches that are closely related to SuperFed. We provide a background of weight shared training in non-federated environments in App. A.

**FL-training challenges.** Our proposed work targets a challenge of efficient deployment of FL-models for on-device inference which is fundamentally different from challenges that occur during training. Specifically, SuperFed aims to reduce the cost of training $k$ global model variants in FL whereas existing works train a single model in FL-training. FedAvg (McMahan et al., 2017) and local SGD (Lin et al., 2020b) train a single model by averaging model updates received from the participating clients. We show that FedAvg is a special case of SuperFed in App. C. Recent modifications to FedAvg like FedDyn (Acar et al., 2021), FedProx (Li et al., 2020), Scaffold (Karimireddy et al., 2021) modify client local training to minimize communication cost for non-i.i.d client datasets. These techniques are complimentary to the proposed work and incorporating them in SuperFed framework is left as future work. Other techniques like Diao et al. (2020); Lin et al. (2020a) enable FL clients to train neural-nets (NNs) that differ in architecture enabling system heterogeneity in FL. SuperFed also allows clients to train architecturally different NNs during training. However, the goal of SuperFed is not to enable system-heterogeneity but satisfy varying deployment scenarios of clients after training. Hence, SuperFed provides k global models to clients during on-device inference, whereas, Diao et al. (2020); Lin et al. (2020a) provide a single global model to every client during inference (post-training). Developing system-heterogeneity aware weight-shared FL training is future work.

**AutoML in FL.** AutoML for federated learning is an emerging field. SuperFed automates the neural architecture selection for varied client deployment scenarios. A known AutoML technique FedNAS (He et al., 2020) automatically searches for NN architecture that maximizes global accuracy in FL. We argue that the goal of SuperFed fundamentally differs from that of FedNAS. FedNAS doesn't target varying deployment scenarios of FL clients whereas SuperFed does. Hence, SuperFed trains k global model variants while FedNAS trains a single global model. SuperFed decouples training from searching of NN architectures whereas FedNAS doesn't. This decoupling is important to allow clients to run local NAS after training in SuperFed. Another line of work FedEx (Khodak et al., 2021) automates hyper-parameter search in FL. Similar to SuperFed, it uses weight-sharing to reduce the communication cost. However, FedEx uses weight-sharing in a different context than that of SuperFed. Specifically, FedEx shares parameter of an NN across different hyper-parameters like learning rate schedules, momentum and weight-decay. Whereas, SuperFed shares parameters of an NN across different NNs that differ in shape and sizes (similar to Cai et al. (2020)). Moreover, FedEx produces a single global model at the end of hyper-parameter optimization in FL. Whereas, SuperFed produces k global models in FL.

**Multi-Model FL.** Another line of FL-research trains multiple models simultaneously but on a fairly different FL scenario than SuperFed. Multi-Model FL (Bhuyan & Moharir, 2022; da Silva et al., 2022) trains $k$ global models (simultaneously) that make predictions on $k$ different tasks (for e.g. one model may predict birds' species while another predicts dogs' species). Whereas, SuperFed trains $k$

| Notation | Explanation |
|----------|-------------|
| $\mathcal{S}$ | The set of clients in FL training, $S = \{1, 2, .., K\}$ |
| $C$ | Client Participation ratio in FL |
| $h$ | Spatial heuristic for subnetwork distribution such that h : $S \to \phi$ |
| $\mathcal{H}(t)$ | Spatio-temporal heuristic for subnetwork distribution such that $\mathcal{H} : \{1, 2, ...T\} \to h$ |
| $\mathcal{M}$(W, arch ,w) | Weight $w$ of subnetwork $arch$ partially replacing weight $W$ of supernetwork |

Table 1: Notations used in Algorithms 1 and 2. Full list of notations are described in Tab. 3.

different global models on a single task (every model in SuperFed makes prediction for the same classes) for varying client deployment scenarios. Multi-Model FL assumes availability of k different datasets on a client device whereas SuperFed trains every model on a single (and same) dataset per client.

## 2 SuperFed: Weight Shared FL Framework

We start by describing our weight shared FL framework in this section. It provides a pluggable algorithmic and implementation base for subnetwork distribution and aggregation heuristics between participating clients—a novel challenge introduced by weight shared FL. We also propose one such SuperFed's heuristic MaxNet which instantiates a specific approach to subnetwork distribution and aggregation. Overall, the goal of weight-shared FL training is to train all subnetworks contained inside supernetwork on all the data partitions. The problem formulation of weight-shared FL is discussed in detail in App. B. Tab. 1 lists some notations used to describe SuperFed.

### 2.1 Federated Setup

Fundamentally, to train a supernetwork in a federated fashion, a server must decide how to distribute its sub-architectures between participating clients. This happens at the beginning of each FL round (line 4 of algorithm 1). The server extracts the subnetworks from the hosted supernet and sends them to the clients (algorithm 1 line 6). Each participating client trains the received subnetwork for a fixed number of local epochs and sends the updated subnetwork parameters back to the server (algorithm 1 line 7). Upon receipt of individual updated subnetwork parameters, the server is then responsible for merging them and assimilating the updates into the shared supernetwork. Critically, it must take into consideration any overlaps or contributions to individual parameters from more than one client (algorithm 1 line 11-15). Note that FedAvg (McMahan et al., 2017) is a special case of algorithm 1 (App. C) where this overlap is constant and equals to the number of clients $K$. However, the overlap cardinality may vary ($\in [1, K]$) in weight shared FL. This is handled on lines 10-16 of algorithm 1. This shared-parameter averaging is one of the key contributions and further investigation opportunity. Now, we describe these two principal framework components in more detail.

### 2.2 Subnetwork Distribution

We start by taxonomizing different subnetwork distribution heuristics on the server side. Fundamentally, the design space for subnetwork distribution heuristics can be divided into two high-level categories: *spatial* and *temporal*.

**Spatial** refers to the distribution of subnetworks among participating clients within a given FL round. Intuitively, this class of heuristics concerns itself with *which* subnets go *where*.

**Temporal** refers to the distribution of subnetworks for an individual client *across* rounds. Intuitively, this class of heuristics concerns itself with *when* a given subnet is forwarded to a given client.

Both spatial (which) and temporal (when) aspects should be taken into consideration for subnetwork distribution for best performance. We propose one concrete spatio-temporal heuristic in this paper (§2.4). Indeed, spatial distribution makes sure that the upper and lower bound of the model family is optimized, while temporal heuristics ensure exposure to all data partitions. The latter can be thought of as a form of temporal load balancing of individual subnetworks across partitioned datasets. To further validate the importance of *spatio-temporal* heuristics, we perform an ablation study, comparing it with random subnetwork distribution (§3.4).

1: Initialize $W$
2: **for** round t = 1,2, ... T **do**
3:     $S_t \leftarrow$ random set of $max(C \cdot K, 1)$ clients ($S_t \subseteq \mathbb{S}$)
4:     $h_t = \mathcal{H}(t)$ // $t^{th}$ spatial subnet distr.
5:     **for** client $k \in S_t$ **do**
6:         $arch_t^k \leftarrow h_t(k)$, $w_t^k \leftarrow \mathcal{G}(W_t, arch_t^k)$
7:         $w_{t+1}^k \leftarrow$ ClientUpdate( k , $w_t^k$ )
8:     **end for**
9:     $W^0 \leftarrow$ zeros($W_t$)
10:     // shared-param avg by overlap cardinality
11:     $W_{sum} \leftarrow \sum_{k \in S_t} n_k * \mathcal{M}(W^0, arch_t^k, w_{t+1}^k)$
12:     $W_{sum} \leftarrow$ replace_zeros($W_{sum}, W_t$)
13:     $W_{cnt} \leftarrow \sum_{k \in S_t} \mathcal{M}(W^0, arch_t^k, n_k * ones(w_{t+1}^k))$
14:     $W_{cnt} \leftarrow$ replace_zeros($W_{cnt}, 1$)
15:     $W_{t+1} \leftarrow \frac{W_{sum}}{W_{cnt}}$
16: **end for**

**Algorithm 1:** Weight Shared FL in Server

1: **Input:** Client j ($j \in S_t$) which has been assigned the largest subnetwork at $t^{th}round$ ($\mathcal{H}(t)(j) = arch_M$), $S_t$ is set of randomly sampled clients $t^{th}$ round
2: $\beta_t = decay(\beta_0, t)$
3: $W^0 \leftarrow$ zeros($W_t$)
4: $W_{sum} \leftarrow \sum_{k \in S_t \setminus j} \frac{(1-\beta_t)}{|S_t|-1} * n_k * \mathcal{M}(W^0, arch_t^k, w_{t+1}^k)$
5: $W_{sum} += \beta_t * n_j * \mathcal{M}(W^0, arch_M, w_{t+1}^j)$ // largest subnet
6: $W_{sum} \leftarrow$ replace_zeros($W_{sum}, W_t$) // no overlaps
7: $W_{cnt} \leftarrow \sum_{k \in S_t \setminus j} \mathcal{M}(W^0, arch_t^k, \frac{(1-\beta_t)}{|S_t|-1} * n_k * ones(w_{t+1}^k))$
8: $W_{cnt} += \beta_t * \mathcal{M}(W^0, arch_M, n_j * ones(w_{t+1}^j))$
9: $W_{cnt} \leftarrow$ replace_zeros($W_{cnt}, 1$) // no overlaps
10: $W_{t+1} \leftarrow \frac{W_{sum}}{W_{cnt}}$

**Algorithm 2:** MaxNet's Shared-Param Averaging For Round $t$ after clients' local updates

## 2.3 SHARED-PARAMETER AVERAGING

Shared parameter averaging is the second principal component of the weight shared FL framework. Fundamentally, it provides a scaffolding for implementing heuristics that aggregate parameter updates from participating clients. The server performs shared-parameter averaging of spatio-temporally distributed subnetworks at the end of each FL round. One naive way of shared-parameter averaging is to keep track of the cardinality of the overlap—the number of overlaps of shared parameters—and average the parameters based on this cardinality (Algorithm 1 line 10-16). We call this simplest FedAvg extension to weight shared supernets as *averaging by overlap cardinality* and analyze it in §3.4. While the framework is designed to accommodate arbitrary spatio-temporally aware aggregation mechanisms, we propose a specific aggregation mechanism in §2.4.

**Limitation.** In order to fully replace previous round's supernetwork parameters in the current round $t$, the largest subnetwork must be trained by at least one client. Note that the weights with zero overlap are kept same as the previous round's weights (algorithm 1 Line 12 and 14). Since largest subnetwork weights $arch_M$ are same as supernetwork's parameters $W$, the largest subnetwork should be included in each round to ensure the cardinality of overlap $\geq 1$. Hence, our framework makes a simplifying assumption (justified in App. J) that a spatio-temporal heuristic will sample the largest network each round for at least one client.

## 2.4 SUPERFED'S WEIGHT SHARED FL TRAINING

We propose a novel spatio-temporal subnet distribution heuristic and shared-parameter averaging in this section. We collectively refer to this combination as MaxNet.

**MaxNet's subnet distribution.** In order to optimize the weight-shared FL objective (App. B equation 1), we first use the sandwich rule (Yu & Huang, 2019b). With the sandwich rule, optimizing the lower and upper bound can implicitly optimize all the architectures in $\phi$. Hence, if $|S_t|$ is the number of clients participating in the $t^{th}$ FL round, then MaxNet heuristic *spatially* samples one smallest, one largest and $|S_t| - 2$ random subnets. However, not every client participates in even single communication round. The heuristic should train the lower and upper bound on as many clients as possible to train muliple model variants. Therefore, *temporally*, MaxNet keeps track of the number of times the smallest and largest subnetworks have been assigned to each client. Hence, in each communication round, the heuristic assigns the smallest and largest subnetworks to the clients which have seen them *the least*. Random subnetworks are assigned to the rest of the clients. We call this subnet distribution as *Tracking-Sandwich*.

**MaxNet's shared-parameter averaging.** The shared-parameter averaging of MaxNet is derived from *averaging by overlap cardinality* described in §2.3. We find that naive way of shared-parameter averaging leads to interference Cai et al. (2020). We report our findings in §3.4. In order to reduce

interference, we propose performing a weighted average among subnetworks (algorithm 2). While averaging the parameters of $|S_t|$ subnetworks, we assign $\beta_t$ ($\beta_t \in R$, $0 < \beta_t < 1$) weight to the largest subnetwork's parameters and $\frac{(1-\beta_t)}{|S_t|-1}$ weight to the parameters of rest of the subnetworks (algorithm 2 line 4-5). The training starts with an initial value $\beta_t = \beta_o$ (say $\beta_o = 0.9$), and $\beta_t$ value is decayed over some communication rounds to reach $\beta_T$ (say $\beta_T = (1/|S_T|)$) (algorithm 2 line 2). Our proposed heuristic has three hyperparameters — 1) initial weight given to maximum subnetwork in averaging $\beta_0$ 2) decay function of $\beta_t$ 3) decay period of $\beta_t$ w.r.t percentage of total communication rounds. We perform a detailed ablation on these hyperparameters in §3.4. We use $\beta_0 = 0.9$ and decay it for $80\%$ of the total communication rounds using cosine decay function as standard hyperparameters for the majority of experiments.

### 2.4.1 REDUCING INTERFERENCE

Interference is a known phenomenon in centralized weight shared training (Cai et al., 2020) in which smaller subnetworks interfere with the training of the larger subnetworks. We demonstrate interference in weight shared federated learning (§3.4) and that weight-shared training in FL context is *non-trivial*. We argue that in order to reduce interference, *it is paramount to facilitate/priortize the training of the largest subnetwork first in the global model family*. Hence, MaxNet weighs the importance of largest subnetwork's parameters and reduces it's influence over time. Specifically, weighted averaging with $\beta_t$ prioritizes the contributions of the largest subnetwork in the earlier phase of training. We later demonstrate that MaxNet's shared-parameter averaging reduces interference among subnetworks (figure 7). In §3.4, we perform a detailed ablation on MaxNet heuristic and its derivations to show attribution of benefit (Tab. 2).

## 3 EXPERIMENTS

We show generality of our proposed algorithm on various federated learning scenarios through comprehensive evaluation. We compare SuperFed's MaxNet and independent training (using FedAvg) on the following dimensions : (1) real-world datasets (2) non-i.i.d degree $\alpha$ (3) client participation ratio $C$ . Our goal is to match FedAvg across diverse FL scenarios and achieve an order-of-magnitude cost reduction w.r.t training a global model family. Furthermore, we also provide a cost analysis for global model family training. Later, we perform a detailed ablation study to highlight key factors which contribute to our results. We run every experiment for three different seeds and report mean and standard deviation.

### 3.1 SETUP

**Baseline.** To our knowledge, there does not exist an FL algorithm which trains a global family of models using weight-sharing. Hence, our baseline is independent SOTA training of DNN subnetworks (without weight sharing) using FedAvg. We call this iFedAvg. While SuperFed trains upto $5 * 10^8$ subnetworks jointly, it is computationally infeasible to train such a large number of subnetworks with FedAvg independently. We train a family of four DNN models using FedAvg (independently) containing both the smallest (lower bound) and the largest (upper bound) subnetworks of the supernetwork (App. F Tab. 4). We compare test accuracy/perplexity of these subnetworks trained using the two approaches and add upto 5 more subnetworks for SuperFed to illustrate that SuperFed trains an order of magnitude more subnetworks which is computationally infeasible for iFedAvg.

**Dataset and Models.** We compare our algorithm with the baseline on four datasets CIFAR10/100 (Krizhevsky et al., 2009), CINIC-10 (Darlow et al., 2018) and PennTreeBank (PTB) (Marcus et al., 1993). For image datasets, The base supernetwork is a ResNet (He et al., 2016) based architecture and contains ResNet-10/26 as the smallest and largest subnetworks respectively. For text dataset, the base supernetwork is a TCN (Bai et al., 2018) based architecture (defined in App. H.1). In SuperFed, we train subnetworks varying in the depth and number of filters in convolution layers (App. F).

**Heterogeneity in Client Data** Similar to (Chen & Chao, 2020; Lin et al., 2020a), we use the Dirichlet distribution for disjoint non-iid training data. The $\alpha$ parameter controls the degree of *heterogeneity* — $\alpha = 100$ is a close to uniform distribution of classes. Lower $\alpha$ values increase per-class differences between data partitions (App. K).

**Local Training Hyper-params.** We keep client's local training hyperparameters same for SuperFed and iFedAvg. For image datasets, our training setting is similar to (Lin et al., 2020a) using local

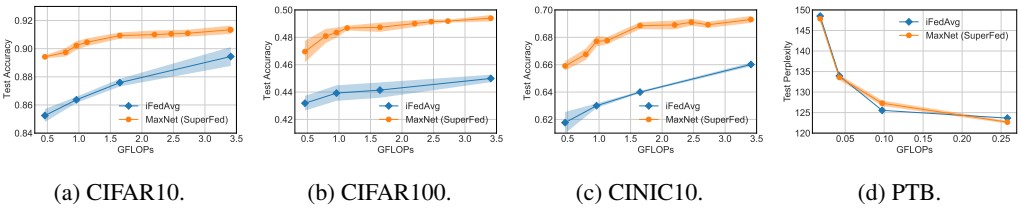

| (a) CIFAR10. | (b) CIFAR100. | (c) CINIC10. | (d) PTB. |

Figure 2: **Datasets**. Test accuracy (higher is better) or perplexity (lower is better) comparison between SuperFed and iFedAvg for different datasets. FL training is done with $C = 0.4$, $\alpha = 100$ and 20/20/100/20 total clients for CIFAR10/100, CINIC-10 and PennTreeBank (PTB) datasets.

SGD (Lin et al., 2020b) with no weight decay and a constant learning rate (no decay). We perform the grid search for learning rate between $\{0.03, 0.1\}$ on training largest subnetwork using FedAvg (App. K). We find the optimal learning rate to be 0.1 and train the model for 5 local epochs for each communication round[1]. We defer local training hyper-params for text dataset to App. H.2.

## 3.2 EVALUATION

We compare our weight shared framework SuperFed with iFedAvg across various federated learning settings in this section.

**Accuracy on Real-world Datasets.** figure 2 compares MaxNet and iFedAvg on four datasets. For this experiment, we keep $C = 0.4$ and $\alpha = 100$. CINIC10/CIFAR10/100 and PTB experiments are run for $R = 1000/1500/2000/100$ communication rounds with total clients as $100/20/20/20$ respectively.
*Takeaway*. SuperFed's MaxNet achieves at par (or better) test accuracy/perplexity (for all subnetworks) than iFedAvg. MaxNet gets closer to iFedAvg as the dataset becomes harder (CIFAR100 is a harder dataset than CIFAR10 Krizhevsky et al. (2009)).

**Effect of Non-i.i.d Degree** $\alpha$. figure 3 evaluates MaxNet as the degree of heterogeneity increases. In this experiment, we use CIFAR10 dataset with $C = 0.4$ and run different $\alpha = \{0.1, 1, 100\}$ settings for $R = 1500/1500/2500$ communications rounds respectively.
*Takeaway*. SuperFed's MaxNet is robust to non-i.i.d-ness and experiences no loss of accuracy relative to iFedAvg across all the $\alpha$ settings. As the degree of non-i.i.d-ness increases, the difference of the test accuracy between both the approaches decreases.

**Effect of Client Participation Ratio** $C$. figure 4 compares SuperFed's MaxNet and iFedAvg by varying percentage of clients participating in each communication round. The experiment is run using CIFAR10 dataset, $\alpha = 100$ and 20 clients. Different $C = \{0.2, 0.4, 0.8\}$ settings are run for $R = 1500/1500/1000$ communication rounds respectively.
*Takeaway*. MaxNet experiences no loss of accuracy relative to iFedAvg across all the $C$ settings as well. MaxNet's tracking makes sure that every client contributes to the global supernet eventually. Hence, MaxNet is resilient to client participation ratios.

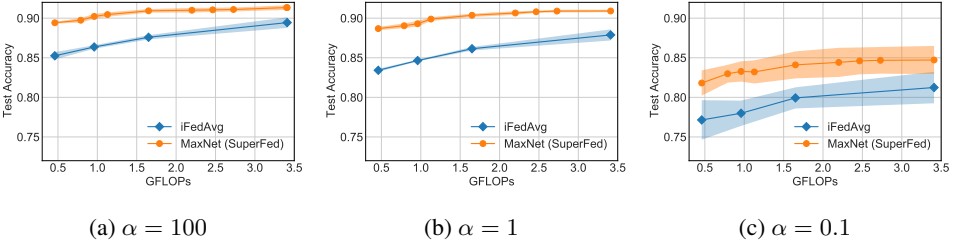

| (a) $\alpha = 100$ | (b) $\alpha = 1$ | (c) $\alpha = 0.1$ |

Figure 3: **Non-i.i.d Degree** ($\alpha$). Test accuracy comparison between SuperFed and iFedAvg for different $\alpha = \{100, 1, 0.1\}$ values run for $1500/1500/2500$ comm. rounds respectively. The lower the $\alpha$ value, the more heterogeneous is the data. Dataset used is CIFAR10 with $C = 0.4$ and 20 clients.

---

[1]We adopt SOTA approach to batchnorm which is known to be problematic in FL non-iid settings (Hsieh et al., 2020)

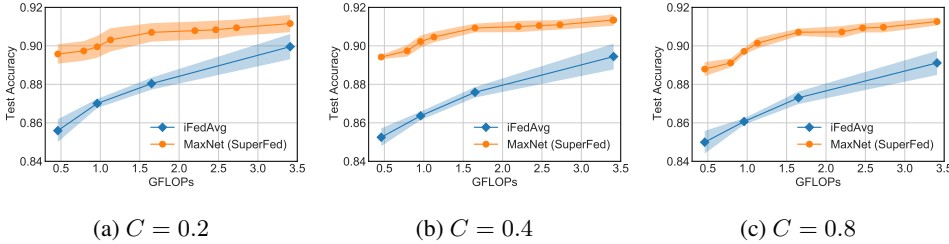

(a) $C = 0.2$      (b) $C = 0.4$      (c) $C = 0.8$

Figure 4: **Client Participation Ratio (C)**. Test accuracy comparison between SuperFed and iFedAvg for $C = \{0.2, 0.4, 0.8\}$ values run for $1500/1500/1000$ FL comm. rounds respectively. Dataset used is CIFAR10 with $\alpha = 100$ and 20 clients.

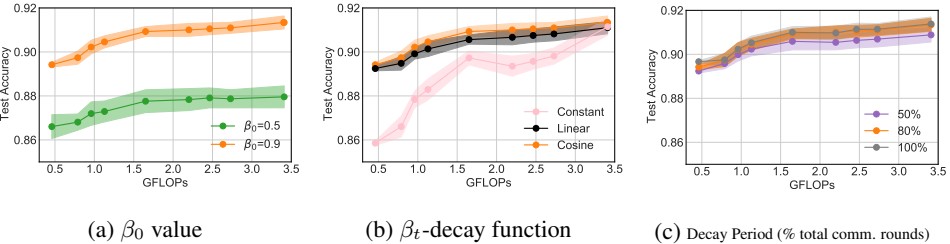

(a) $\beta_0$ value      (b) $\beta_t$-decay function      (c) Decay Period (% total comm. rounds)

Figure 5: **MaxNet's hyperparameters. a)** Initial $\beta_0$ value used in weighted aggregation of subnetwork parameters. $\beta_t$ is assigned to maximum subnetwork and $(1 - \beta_t)$ to the rest **b)** For decaying functions (except constant), $\beta_t$ is decayed from $0.9 \rightarrow$ uniform within the same number of rounds (80% of total rounds) **c)** Decay period is defined as % of total rounds until which $\beta_t$ decay occurs.

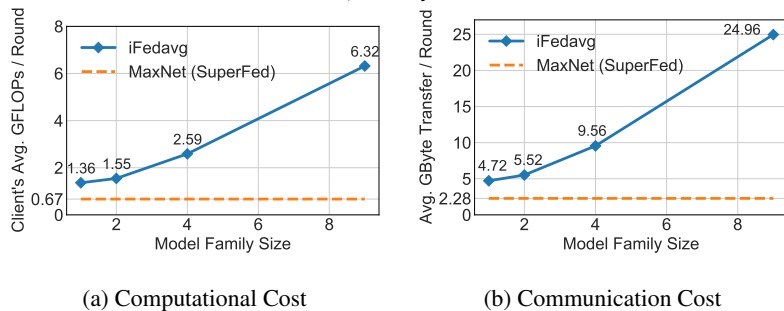

(a) Computational Cost      (b) Communication Cost

Figure 6: **Costs of training a model family in FL.** Costs of training a model family with varied size is shown. Blue line represents cost of training each subnetwork in the family using FedAvg. The dotted line (orange) line represents costs of training a model family of size $\approx 5 * 10^8$ using SuperFed. Cost calculation details are described in App. G.

## 3.3 COST COMPARISON

We now compare SuperFed and iFedAvg w.r.t computational and communication costs for training a family of models. We defer cost calculation details to App. G.

*Takeaway.* figure 6 shows computational and communication costs of iFedAvg as the model family size increases. The dotted line represents cost for training $\approx 5 * 10^8$ subnetworks using SuperFed's MaxNet. Clearly, as the size of model family increases the cost of training model family with iFedAvg approach increases. In fact, training a family of 9 models with iFedAvg is 9.43x costlier computationally and 10.94x costlier in communication than training $\approx 5 * 10^8$ subnetworks using SuperFed's MaxNet. Hence, iFedAvg is *infeasible* if used for training subnetworks of the same order of magnitude as SuperFed. Furthermore, we emphasize that SuperFed incurs smaller cost than independently training the largest network with iFedAvg (model family size as 1, App. E), making it the preferred method of federated training for any family of models, however small.

## 3.4 ABLATION STUDY

We now study attribution of benefits by performing an ablation on 1) Spatio Temporal Heuristics 2) Hyperparameters of MaxNet heuristic.

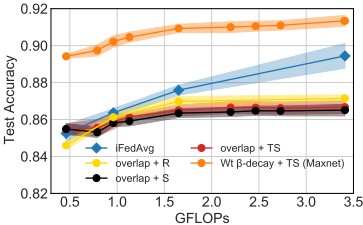

Figure 7: **SuperFed's Spatio-Temporal Heuristics.** Test accuracy comparison for various spatio-temporal heuristics. The dataset used is CIFAR10 with $C = 0.4$, $\alpha = 100$ and 20 clients.

Table 2: Description of subnetwork distribution and shared-parameter averaging for different spatio-temporal subnetwork distributions in SuperFed

| Heuristic | Subnetwork Dist | Averaging |
|---|---|---|
| overlap + R | Random | overlap cardinality |
| overlap + S | Sandwich | overlap cardinality |
| overlap + TS | Tracking-Sandwich | overlap cardinality |
| Wt $\beta$-decay + TS (MaxNet) | Tracking-Sandwich | Wt $\beta$-decay + overlap cardinality |

**Spatio Temporal Heuristics.** Tab. 2 lists the spatio-temporal heuristics tried as a part of this ablation study. For fair comparison, all the training hyperparameters are kept constant — CIFAR10 dataset, $C = 0.4$ and $\alpha = 100$. figure 7 compares the test accuracy of all the spatio-temporal heuristics.
*Takeaway.* First, MaxNet has the best accuracy for all the subnetworks among all other heuristics. Second, notice that the simple *overlap cardinality* averaging (overlap + R) with random subnetwork distribution *underperforms* for all subnetworks compared to independently training them. While, both sandwich (overlap + S) and tracking-sandwich (overlap + TS) subnetwork distribution improve smallest subnetwork's accuracy as compared to random subnetwork distribution (overlap + R) and match the accuracy of independently training it. Tracking-sandwich (overlap + TS) performs slightly better than sandwich (overlap + S). However, the largest subnetwork's accuracy is still sub-par compared to iFedAvg. We attribute the reason for sub-par accuracy of the largest subnetwork to interference (for overlap + (R,S,TS) in figure 7), i.e. smaller subnetworks interfere with the training of the larger subnetworks Cai et al. (2020). MaxNet prioritizes contributions of the largest subnetwork in the earlier phase of training and decays its importance over time. This reduces interference and achieves no loss of accuracy compared to independently trained subnetworks.

**Hyperparameters of MaxNet.** MaxNet's hyperparameters are described in §2.4. We perform a grid search on the following hyperparameters : $\beta_0 : \{0.9, 0.5\}$ , decay function {linear, cosine, constant}, decay period {50%, 80%}. figure 5 compares test accuracy of the subnetworks for different hyperparameters.
*Takeaway. Hyperparameter $\beta_0$ has a major effect on test accuracy* of subnetworks (figure 5a). Starting the training with larger $\beta_0$ value helps reduce interference. Hence, training of the largest subnetwork is preferred more in the earlier phase of training, which also benefits the smaller networks. Also, keeping $\beta_t = 0.9 = const$ only improves largest subnetwork's accuracy. Decaying $\beta_t$ over communication rounds increases accuracy for smaller subnetworks without reducing the largest subnetwork accuracy(figure 5b). Similar empirical evidence of the benefit of gradually reducing larger subnetwork's importance in training is also seen in OFA (Cai et al., 2020). Moreover, cosine decay performs better than linear decay. Longer decay period (80% of the total comm. rounds) further helps in achieving better test accuracy (figure 5c).

## 4 CONCLUSION

SuperFed is the new framework for weight shared FL. Its pluggable architecture is used to evaluate a newly proposed set of distributed training algorithms for spatio-temporal subnetwork distribution and model parameter aggregation, collectively called MaxNet. MaxNet is able to train multiple DNNs jointly with weight sharing in a federated fashion, matching state-of-the-art FedAvg w.r.t accuracy. SuperFed amortizes the cost of training over an arbitrary selection of subnetworks, while conventional FL techniques (e.g., FedAvg) incur $O(k)$ training cost for any selection of $k$ models from the "family of models". SuperFed opens new directions for future work: adjusting MaxNet for client resource constraints (e.g., compute, bandwidth, memory capacity) and developing theoretical insights on training enabled by MaxNet. We believe that SuperFed takes an important first step towards co-training model families in a federated fashion cost-efficiently, achieving an order of magnitude communication and computation cost reduction (even with as few as $k = 9$ models) while maintaining accuracy through interference mitigation.

## 5 REPRODUCIBILITY STATEMENT

We provide anonymized source code `https://anonymous.4open.science/r/SuperFed/` for the experimental results included in the main text. The code includes full instructions to perform weight-shared federated learning on the datasets and settings considered in the paper.

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

## A    BACKGROUND ON WEIGHT-SHARED TRAINING

OFA (Cai et al., 2020) proposes a progressive shrinking method for weight-shared training of $10^{19}$ subnetworks jointly by using a neural supernetwork. The training is facilitated by a pre-trained teacher network which is used for distillation throughout the training. Training a supernetwork naively causes "interference" (Cai et al., 2020). Hence, OFA takes a phased approach to training, by training larger subnetworks first and progressively shrinking the dimensions of a subnetwork being trained. BigNAS (Yu et al., 2020) proposes a one-shot approach (doesn't use any pretrained network) for training a supernetwork using inplace distillation (Yu & Huang, 2019a). It also proposes several training hyperparameters which optimize the supernetwork training. However, these weight-shared training techniques are proposed for centralised setting with i.i.d data. The proposed SuperFed is a one-shot weight shared training framework for non-i.i.d decentralised data. We empirically show that naive application of FL techniques for globally training weight shared supernetwork structures causes interference among subnetworks in non-i.i.d setting (§3.4).

## B    PROBLEM FORMULATION

Now, we formulate the objective for our weight shared federated learning (notations listed in App. D Tab. 3). Later, we discuss our proposed heuristic to minimize the objective. In FedAvg (McMahan et al., 2017), a single global model with parameters $w$ is trained by minimizing the following objective function - $\min_w \sum_{k=1}^{K} \frac{n_k}{n} * F_k(w)$. where there are $K$ clients with their own data partition $P_k$ such that the size of the partition is denoted by $n_k$ ( where $n = \sum_{k=1}^{K} n_k$ is the total number of data points). $F_k(W) = \sum_{i \in P_k} f_i(w)$ denotes the loss for data points of partition $P_k$

Meanwhile, in weight shared training, OFA's goal is to train every possible architecture in the supernetwork. Assume $\phi : \{arch_1 , arch_2 , ... arch_M \}$ as a DNN model family with $arch_M$ being the largest subnetwork. The shared weights of these architectures are denoted by $W$ (supernet's parameters). Hence the objective of OFA (Cai et al., 2020) can be formalized as follows: $\min_W \sum_{arch_i \in \phi} L(\mathcal{G}(W, arch_i)))$. where $w = \mathcal{G}(W, arch_i)$ denotes a selection (subset) of $arch_i$ parameters from shared weights $W$. $L(w)$ denotes the loss of subnetwork's parameters $w$ ($w \in W$) on a single central dataset.

As stated in §1, the goal of this work is to train a global model family of size $|\phi|$. Thus, the objective function of such training takes the following form:

$$\min_W \sum_{arch_i \in \phi} \sum_{k=1}^{K} \frac{n_k}{n} * F_k(\mathcal{G}(W, arch_i)) \tag{1}$$

Indeed, the goal of weight shared FL training is to train all the architectures in the family $\phi$ on all the data partitions $\{P_1, P_2, ..., P_K\}$. Weight shared FL aims at training a family of models $\phi$ at an order of magnitude lesser training cost (both communication and computational) relative to independently training these networks in FL while matching or even surpassing accuracy. We compare weight shared FL trained subnetworks with independently FL trained subnetworks in §3.2.

## C    FEDAVG: A SPECIAL CASE OF SUPERFED

FedAvg (McMahan et al., 2017) is a special case of SuperFed's algorithm (algorithm 1). When the family of models contains only a single subnetwork i.e. $\phi = \{arch\}$, then algorithm 1 only trains the weights of that subnetwork. The spatio-temporal heuristic $\mathcal{H}$ becomes $\mathcal{H}(t)(k) = arch \ \forall t \in \{1, 2, ..T\}, k \in \mathcal{S}$. Since the same subnetwork is sent to all the clients participating in a round, the number of overlaps for weights of subnetwork $arch$ equals the number of participating clients $|S_t|$. Hence the averaging method reduces to FedAvg's averaging mechanism for subnetwork $arch$.

# D  DESCRIPTION OF NOTATIONS

Table 3: Notations used in the paper

| Notation | Explanation |
|---|---|
| $K$ | The number of clients |
| $\mathcal{S}$ | The set of clients involved in FL training, $S = \{1, 2, .., K\}$ |
| $T$ | The total number of communication rounds in |
| $P_k$ | Dataset partition for client $k$ |
| $F_k(w)$ | Sum of loss of data points in $P_k$ for DNN weight $w$ |
| $n_k$ | Size of dataset partition $P_k$ ($|P_k|$) |
| $C$ | Client Participation ratio in federated setting |
| $\phi$ | Family of DNN models or a set of DNN architectures |
| $W$ | Weight of supernetwork |
| $w$ | Weight of subnetwork contained inside supernetwork |
| $arch$ | DNN architecture (subnetwork) such that $arch \in \phi$ |
| $\mathcal{G}(W, arch)$ | Selection of weights for DNN $arch$ from supernet weights $W$ |
| $arch_M$ | largest subnetwork in $\phi$ such that $\mathcal{G}(W, arch_M) = w_M = W$ |
| $h$ | Spatial heuristic for subnetwork distribution such that h : $S \to \phi$ |
| $\mathcal{H}(t)$ | Spatio-temporal heuristic for subnetwork distribution such that $\mathcal{H} : \{1, 2, ...T\} \to h$ |
| $\mathcal{M}(W,\ arch,\ w)$ | Weight $w$ of subnetwork $arch$ partially replacing weight $W$ of supernetwork (super-imposition) |

# E  COST COMPARISON FOR TRAINING THE LARGEST SUBNETWORK

Since supernetwork's weights are same as the largest subnetwork's weights, we compare SuperFed and FedAvg w.r.t communication and computational costs of training the same weights for total communication rounds $T$. In SuperFed, other subnetworks (i.e., subsets of the largest subnetwork parameters) are also trained along with the largest subnetwork while FedAvg only trains the largest subnetwork. We assume that the largest subnetwork is assigned to atleast one client for every round in SuperFed (assumption in §2.3). Note that the overall time spent by a client in training the neural network(s) send by the server over multiple rounds would be less in SuperFed than FedAvg. Moreover, the amount of bytes exchanged between the server and clients would be less in SuperFed than FedAvg in every round. This is because in SuperFed, smaller subnetworks are also sent to some clients other than just sending the largest subnetwork to every client (FedAvg). These smaller subnetworks not only take less training time (computational) but also have less parameters (communication). Hence, our proposed weight shared federated learning framework trains a family of models with lesser computational and communication costs than training the largest subnetwork using FedAvg.

# F  DNN ARCHITECTURE SPACE AND SELECTION FOR IMAGE DATASETS

We modified OFAResNets — a supernetwork proposed in OFA (Cai et al., 2020) to have smaller size (10-26 layers) models and make it amenable to $32x32$ image size. A typical ResNet architecture He et al. (2016) has multiple stages and each stage contains a list of residual blocks. We implement the supernetwork with 4 such stages. Next, we describe the two elastic dimensions we use to vary subnetworks in size.

**Depth.** It decides the no. of residual blocks per stage. Hence, this dimension is usually specified using a list of size 4 with each value denoting the depth of that stage. Our smallest subnetwork keeps a depth of two per stage. Hence these two per-stage residual blocks are shared among every subnetwork. Depth elastic dimension can also be specified using a single integer say $d$ which denotes $d$ depth for every stage in the supernetwork.

**Elastic ratio.** This dimension changes number of filters and channels of convolution layers in a residual block. The block has two convolutional layers with input, middle and output channels. Elastic ratio is a ratio of output to middle channel only affecting the number middle channels in the block. The elastic ratio is specified via list of size 12 expressing number of middle channels for each residual block. It can also be specified using a single integer say $e$ which denotes $e$ elastic ratio for

every residual block in the supernetwork.

Tab. 4 lists the subnetworks selected for comparing SuperFed with iFedAvg. Out of the 9 subnetworks, 4 of them are also trained using FedAvg independently. SuperFed $\approx 5 * 10^8$ subnetworks including the nine subnetworks listed in Tab. 4.

Table 4: Dimensions of subnetwork architectures shown in all the figures. Out of nine architectures, four of them are also trained using FedAvg independently for comparison.

| Subnetwork Depth | Subnetwork Expand Ratio | GB | FedAvg Model Family |
|---|---|---|---|
| [0,0,0,0] | 0.1 | 0.40 | ✓ |
| [0,0,0,1] | 0.14 | 0.83 | |
| [0,1,0,1] | 0.14 | 0.85 | ✓ |
| [0,1,1,1] | 0.14 | 0.94 | |
| [1,1,1,1] | 0.18 | 1.17 | ✓ |
| [1,1,1,2] | 0.22 | 1.91 | |
| [1,2,1,2] | 0.22 | 1.94 | |
| [1,2,2,2] | 0.22 | 2.08 | |
| [2,2,2,2] | 0.25 | 2.36 | ✓ |

## G  COST CALCULATION

### G.1  COMPUTATIONAL COST

We define computational cost as the average time spent/computations done by the client in FL training. Hence, we estimate average computational cost in a single FL communication round. Note that the cost is proportional to the GFLOPs of the model (subnetwork) that it is training. Now, we describe cost calculation for the two approaches compared in §3.

**SuperFed.** Sum of GFLOPs of all the subnetworks that each clients sees over the course of training procedure is done. Then, an average is calculated over the total number of clients and total number of communication rounds.

**iFedAvg.** Here all the subnetworks of a model family are trained independently. The GFLOPs of a subnetwork is proportional to the time client spends per round in FedAvg training . Hence, the sum of GFLOPs of all subnetworks in the family is the average computational cost per round in iFedAvg.

### G.2  COMMUNICATION COST

Communication cost is defined by the average bytes transferred between the clients and the server in FL training. The total communication cost is the sum of communication cost per round. Hence, we first calculate average communication cost per FL round. The average includes both the download and upload of the model. The bytes transferred to the clients and vice-versa depend on the size of the subnetwork. Now we describe communication cost calculation -

**SuperFed.** The sum of size (in GB) of subnetworks distributed to each client at each communication round is calculated. Then average is taken over total communication round.

**iFedAvg.** We first calculate the average communicaton cost per round for training a single subnetwork (say $arch_i$). This is typically given by - $2 * |S_t| * \text{sizeOf}(arch_i)$ ($|S_t|$ denotes number of participating clients). Since every subnetwork is trained independently the total communication cost per round is given by - $2 * |S_t| * \sum_{arch_i \in \phi} \text{sizeOf}(arch_i)$. In figure 6, we show costs for training a model family of size upto nine subnetworks with iFedAvg.

## H  DNN ARCHITECTURE SPACE AND CLIENT LOCAL TRAINING FOR TEXT DATASET

In this section, we describe our TCN Bai et al. (2018) supernetwork and local training hyperparameters associated with the text dataset experiment.

## H.1 SUPERNET BASED ON TCNS

A standard TCN architecture (Bai et al., 2018) for PTB dataset Marcus et al. (1993) consists of eight layer (or four temporal blocks) with 600 input and output channels. We enable two elastic dimensions in the standard TCN architecture — depth and expand ratio. Depth defines the the number of blocks to be activated whereas the expand ratio defines the number of output channels of the first layer of each block. Overall, the depth choices considered are {0,1,2} with 0 denoting activating top two blocks only and expand ratio choices are {0.1, 0.2, 0.5, 1.0}.

## H.2 LOCAL TRAINING HYPER-PARAMS

For client local training in text dataset, the initial learning rate is kept as 4 which is a standard learning rate to train TCN on PTB dataset. We decay the learning rate every 50 local epochs by 0.1. We use gradient norm clipping as 0.35 and batch size 16. We run five local epochs per client for every communication round

## I NEURAL ARCHITECTURE SEARCH

Table 5: NAS performed for 3 GFLOPs constraints using seed 0 best model checkpoint for cifar10 alpha 100.

| NAS Constraint (GFLOPs) | Test Accuracy | GFLOPs | Subnetwork Architecture |
|---|---|---|---|
| 1.0 | 89.98 | 0.94 | d:[1,0,0,0], e:[0.14, 0.18, 0.22, 0.1, 0.18, 0.14, 0.22, 0.18, 0.1, 0.22, 0.22, 0.14] |
| 2.0 | 91.25 | 1.99 | d:[1,2,2,0], e:[0.25, 0.22, 0.22, 0.18, 0.18, 0.18, 0.22, 0.18, 0.18, 0.18, 0.25, 0.18] |
| 3.0 | 91.53 | 2.72 | d:[2,2,2,2], e:[0.14, 0.22, 0.18, 0.25, 0.22, 0.14, 0.18, 0.25, 0.25, 0.25, 0.18, 0.1] |

The goal of weight shared FL is to offer a rich latency accuracy trade-off space so that clients can choose one or more models for her deployment needs. In this section, we demonstrate that once the supernetwork is trained, a neural architectural search (NAS) can be performed by the clients to extract Pareto-optimal subnetworks. Similar to Sahni et al. (2021), we use evolutionary algorithm to search for pareto-optimal subnetworks for a given FLOP constraint. Tab. 5 lists different subnetworks found by the search algorithm for different GFLOPs constraints. The subnetworks are extracted from supernetwork trained with the following setting - $C = 0.4$, $\alpha = 100$, 20 clients and CIFAR-10 dataset. The subnetworks are extracted *without retraining*.

## J JUSTIFICATION OF SUPERFED'S LIMITATION

We first highlight that goal of SuperFed is to target multiple deployment scenarios of clients rather than assume system-heterogeneity during training. Furthermore, we argue that resources are extremely dynamic and scarce in deployment (discussed in §1) than during FL-training. One of the foremost requirement of FL is to make the model training invisible to the user i.e. it should not slow down the smartphone or drain its battery (Bonawitz et al., 2019). Hence, smartphones usually participate in FL-training when they are idle, above a certain battery level and connected to unmetered network (Kairouz et al., 2021). Since resources are fairly available at FL-training time, it is reasonable to assume that atleast one client takes the largest subnetwork during FL-training in a given comm. round (a limitation discussed in §2.3).

## K LEARNING RATE GRID SEARCH

We perform a grid search on learning rate for the largest subnetwork in the model family using FedAvg. figure 8 compares learning rates 0.03 and 0.1. This shows that learning rate 0.1 achieves a better test accuracy for 1500 communication round. To have less complexity, we keep the same learning rate for training each subnetwork using FedAvg. SuperFed also uses learning rate as 0.1 for fair comparison.

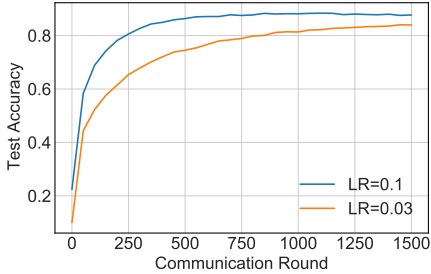

Figure 8: **LR Grid Search.** Comparison of test accuracy for two different learning rate values {0.03, 0.1} on CIFAR-10 dataset using FedAvg. The total number of clients are 20 with $C = 0.4$ and $\alpha = 100$.

## L    CLIENT CLASS DISTRIBUTION

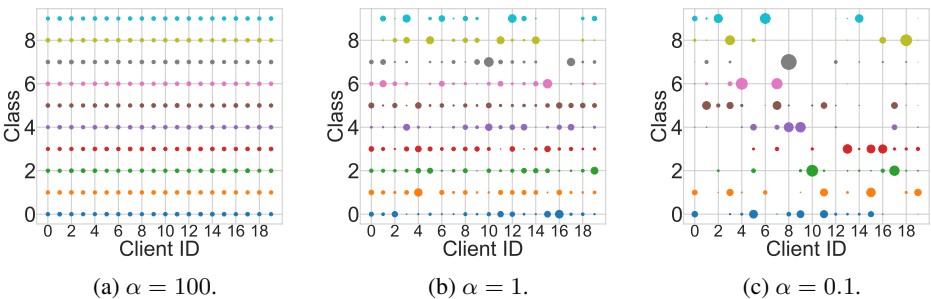

(a) $\alpha = 100$.          (b) $\alpha = 1$.          (c) $\alpha = 0.1$.

Figure 9: **Non-i.i.d Degree** ($\alpha$). Client class distribution visualization across different $\alpha = \{100, 1, 0.1\}$ values. Showing visualizations for CIFAR10 dataset partitioned using Drichlet distribution among 20 clients.

figure 9 visualizes distribution of classes among 20 clients. The size of dots symbolizes the amount of data points for that particular class in client's dataset partition. Partition is done using dirichlet distribution which is parameterized by $\alpha$. As seen in the figure, the lower the $\alpha$ value the higher the degree of non-i.i.d-ness. Specifically for $\alpha = 0.1$, many clients don't have even a single data point for some classes.

