# OpenReview forum: "SuperFed: Weight Shared Federated Learning"
_ICLR.cc/2023/Conference — Submitted to ICLR 2023_

### Official Review · Reviewer_uEYf · 2022-10-19

**Confidence:** 3
**Correctness:** 3
**Technical Novelty And Significance:** 3
**Empirical Novelty And Significance:** 3
**Recommendation:** 5

**Clarity, Quality, Novelty And Reproducibility:**

The clarity of this paper is not so satisfactory. The technical quality, novelty, and reproducibility are moderate.

**Strength And Weaknesses:**

Strengths:
1. The problem studied in this paper is interesting.
2. The proposed method seems to be effective.
3. The concept of temporal-spatial subnetwork distribution is interesting.

Weaknesses:
1. The relations between this work and existing client-heterogeneous FL works [1-3] are not clarified ([2] and [3] are also not included in this paper). The authors mention that "Diao et al. (2020); Lin et al. (2020a) provide a single global model to every client", which is quite confusing because Diao et al. (2020) distribute different models to different clients. In fact, these methods can choose to distribute models in different sizes to different clients in different rounds. Moreover, [1] and [3] also consider weight sharing mechanisms. Thus, it is not clear what is the difference between the scenario in this paper and existing works, as well as the unique advantage of the proposed method.
2. The authors state that "To our knowledge, there does not exist an FL algorithm which trains a global family of
models using weight-sharing". This may not be the case. Thus, more comparisons with existing baselines are needed.
3. The methodology section is somewhat too brief, and many details are not well clarified.


[1] HeteroFL: Computation and communication efficient federated learning for heterogeneous clients. arXiv preprint arXiv:2010.01264.
[2] FedHM: Efficient Federated Learning for Heterogeneous Models via Low-rank Factorization. arXiv preprint arXiv:2111.14655.
[3] No One Left Behind: Inclusive Federated Learning over Heterogeneous Devices. KDD 2022.

**Summary Of The Paper:**

This paper introduces a federated learning method that supports training multiple models in various sizes simultaneously without heavy costs and performance loss. The core idea is weight sharing, where different models can share common knowledge. Experimental results show the effectiveness of the proposed method.

**Summary Of The Review:**

This work is interesting but its insufficient clarity is a major concern. The technical contribution seems to be not very big. Thus, my recommendation is "marginally below the acceptance threshold".

---

> ### Author Response · Authors · 2022-11-12
> **Response to Reviewer  uEYf**
>
> We appreciate that the reviewer finds our work effective and interesting. Thank you for pointing out a typo, we update the sentence in the paper revision to “"Diao et al. (2020); Lin et al. (2020a) provide a single global model to every client during inference (post-training)"”. We further provide clarifications to the questions raised by the reviewer -
> - **The Difference in Goals between SuperFed and HeteroFL/FedHM/InclusiveFL.** The goal of SuperFed is to train a family of models for post-training inference under dynamic on-device deployment conditions. This is significantly different from the cited papers that focus on tackling resource constraints during training itself. The difference in the goal is stated in Sec1.
> - **Different Final Output of SuperFed and HeteroFL/FedHM/InclusiveFL.** Even though the reviewer correctly states that cited works send different models to different clients at training time (also mentioned in Sec1.1 for heteroFL), the output of the whole training procedure is still a single global model for post training inference . This is in contrast to our goal of producing a family of models as a direct output of the proposed training procedure.
>   - We would like to quote a line from the abstract of heteroFL paper(https://arxiv.org/abs/2010.01264) to validate our claim : “Our solution can enable the training of heterogeneous local models with varying computation complexities and still produce a single global inference model”
> - **Different behavior of SuperFed and HeteroFL/FedHM/InclusiveFL in our setting.** Cited works can send different models to different clients, however, we assert that this will not happen in the absence of resource constraints during training (the setting considered in SuperFed and stated in Sec1.1 and Appendix I).
>   - Note that the cited works will always send the largest subnetwork to each client if its resources can train the largest subnetwork (i.e., in the case of no resource constraints). Therefore, the cited works will end up training only the largest subnetwork in the absence of resource constraints. Instead, Maxnet assigns different subnets spatio-temporally sampled to each client using its proposed subnet distribution that leads to training a global family of models for efficient on-device inference post-training
>
> Lastly, we emphasize that considering resource constraints at training time in the SuperFed framework is left for future work (stated in Sec 1.1 and Sec 4). The goal, then, will be to train a global family of models in a resource constraint environment AND produce a family of models at the same time.

---

> > ### Author Response · Authors · 2022-11-18
> > **after addressing the key reviewer's concern, are there any lingering doubts?**
> >
> > Thanks again for reviewing this paper and your service to the community. We believe we've directly addressed your key concern w.r.t. heterogeneous FL. Today is the last day of the open response period. Could you please let us know if you have any follow up questions or lingering doubts? That would be very helpful if we were given the opportunity to directly and openly address them as well. Thanks!

---

> > > ### Author Response · Authors · 2022-11-22
> > > **have we satisfactorily addressed the heteroFL confusion?**
> > >
> > > I would like to directly quote this paragraph from the intro, p2. We'd like to call the reviewer's attention to explicit differentiation from "existing FL approaches" (producing a single model). Does this not directly address the reviewer's confusion about heteroFL?
> > >
> > > _In contrast to existing FL approaches that produce a single model, we need to produce multiple model variants (varying in size/latency) for efficient on-device inference. However, training these model variants independently is computationally prohibitive (Cai et al., 2020). This is particularly true for FL, where training these variants independently will cumulatively inflate the communication cost as well as the computation cost. Thus, it is imperative to develop techniques for training multiple models in a federated fashion cost efficiently without any accuracy loss—achieving asymptotic cost improvements relative to independently training them._ (Section 1, page 2, para 3)

---

### Official Review · Reviewer_8Rfx · 2022-10-25

**Confidence:** 4
**Correctness:** 2
**Technical Novelty And Significance:** 3
**Empirical Novelty And Significance:** 3
**Recommendation:** 5

**Clarity, Quality, Novelty And Reproducibility:**

Clarity: reasonable.
Quality: the experimental evaluation remains limited to similar datasets that are the same or closely related to the non-federated variants of the work.
Novelty: while there has been some work on hyper parameter tuning in FL, to my knowledge the work introduces the first federated OFA method.
Reproducibility: it is unclear if code will be provided.

**Strength And Weaknesses:**

### Strengths:
1. The problem setting is well-motivated and a practical algorithm here would be valuable for applications of federated learning.
2. The authors conduct ablation studies of their solutions for handling interference due to weight-sharing.
3. Experimentally there is a comparison to reasonable baselines (iFedAvg) and the authors demonstrate improvement by the method in obtaining a superior accuracy-complexity tradeoff. There is also evaluation of the method at different heterogeneity levels, albeit synthetically generated via classes.

### Weaknesses:
1. The main weakness of the paper remains the limited evaluation. The proposed algorithm is evaluated on three closely related datasets, all containing natural images with the same size. Most FL papers will at least include a text application alongside evaluation on image data, given that text is one of the dominant applications of FL. There is not a strong reason to know that the proposed method will directly extend beyond image data, especially since the non-FL work this paper is based on (OFA) focuses on vision. I believe to be impactful the paper would at least have to evaluate on one of the text datasets in LEAF (Caldas et al., 2019), or on more recent datasets based on Reddit or StackOverflow.
2. It is unclear if code will be provided for reproducibility.

### References:
1. Caldas, Duddu, Wu, Li, Konecny, McMahan, Smith, Talwalkar. LEAF: A benchmark for federated settings. arXiv 2018.

**Summary Of The Paper:**

This paper introduces a weight-sharing approach for simultaneously training multiple models with different architectures to satisfy different deployment needs in a federated setting. Their method combines existing techniques for weight-sharing with new approaches for distributing architectures and averaging parameters. The resulting algorithm is evaluated on three vision datasets. Note that I have reviewed this paper before and parts of my review are taken from my previous evaluation.

**Summary Of The Review:**

The problem addressed is well-motivated and the authors obtain reasonable empirical results on the datasets considered, but the evaluation remains limited to CIFAR-like datasets, which makes the applicability to practical FL unclear.

---

> ### Author Response · Authors · 2022-11-12
> **Response to Reviewer  8Rfx**
>
> Thank you for your valuable input. To address your concern, we add a text dataset experiment in the paper revision (Fig 2d). The experiment evaluates SuperFed (MaxNet) on PennTreeBank (PTB) text dataset and new NN architecture based on temporal convolution networks (https://arxiv.org/abs/1803.01271). The experiment demonstrates the generality of our proposed approach, as SuperFed (Maxnet) matches the test perplexity of independently trained networks using FedAvg. The experiment is conducted on 20 clients with client participation ratio as 0.4.
>
> We have also made available our anonymized source code (https://anonymous.4open.science/r/SuperFed/) with instructions on performing federated weight-shared training and reproducing our experiments. We plan to open source this code when the paper is accepted.

---

> > ### Author Response · Authors · 2022-11-18
> > **SuperFed generalizes to text datasets -- any other lingering concerns we could directly address?**
> >
> > Thanks again for reviewing this paper and your service to the community.
> > We believe we've directly addressed your primary concern w.r.t generalizability of the proposed techniques to text datasets.
> >
> > We conducted extensive experiments and confirmed that, indeed, our proposed work generalizes.
> >
> > Could you please let us know if you have any follow up questions or lingering doubts? It would be very helpful if we were given the opportunity to directly and openly address them as well.
> >
> > Thank you so much for your feedback and consideration.

---

> > > ### Comment · Reviewer_8Rfx · 2022-11-20
> > > **Response**
> > >
> > > Thank you for the update. I have some questions concerning the experiments:
> > > 1. It seems a stretch to say the results generalize to PTB given that its improvement seems much smaller than on the vision datasets, and it is sometimes worse than iFedAvg (e.g. at .10 GFLOPs). Is the only advantage for the PTB case then that SuperFed effectively has many more models than iFedAvg?
> > > 2. What was the motivation behind the choice of dataset? PTB is a somewhat disappointing choice as it is not often used in FL, and the dataset is non-i.i.d. As it stands, the non-i.i.d. evaluation in the paper is somewhat limited, and standard text datasets such as the ones I mentioned in the review offer natural, non-synthetic forms of client heterogeneity.

---

> > > > ### Author Response · Authors · 2022-11-20
> > > > **Response to  Reviewer 8Rfx**
> > > >
> > > > We thank the reviewer for following up on our paper revision. We clarify the reviewer's concerns -
> > > > 1. **Goal of SuperFed.** The goal of SuperFed is to match test accuracy/perplexity of independently trained networks (using FedAvg) at a fraction of training cost, by achieving asymptotic cost of O(1) for training k models. The advantage of using SuperFed is that the training cost is order of magnitude (upto $9x$) less than independently training the models (Fig 6) while matching accuracy/perplexity.
> > > > 2. **Non-iid Experiments exist: Fig 3**. We point the reviewer to Fig3 where we evaluate SuperFed extensively on different levels of non-i.i.dness in the data. Specifically, \alpha=0.1 setting is extremely heterogeneous wrt class labels and SuperFed is shown to match accuracy of independently trained networks using FedAvg.
> > > > 3. **Dataset Choice.** The dataset was chosen because the open source implementation of Temporal Convolution Networks (https://github.com/locuslab/TCN) evaluates the network on PTB dataset. To the best of our knowledge, there doesn't exist an open source implementation of supernetwork for text dataset which is why we implement our own supernet based on TCN architecture that can be evaluated on PTB text dataset. We also highlight that developing supernetworks based on architectures like BERT (network that is used against the datasets pointed out by the reviewer) is an open research question and out of the scope of this work.

---

> > > > > ### Author Response · Authors · 2022-11-22
> > > > > **goal: training cost reduction without accuracy loss (references to the text)**
> > > > >
> > > > > I would like to draw the reviewer's attention to Section 1, page 2, para 3, quoted below. We explicitly state here, as well as in the abstract, that the intention is NOT to increase accuracy. The goal is to achieve both asymptotic and empirical training cost reduction without sacrificing accuracy. We show that we achieve this. From the paragraph below, the key sentence is **``Thus, it is imperative to develop techniques for training multiple models in a federated fashion cost efficiently without any accuracy loss—achieving asymptotic cost improvements relative to independently training them.''**
> > > > >
> > > > > _In contrast to existing FL approaches that produce a single model, we need to produce multiple model variants (varying in size/latency) for efficient on-device inference. However, training these model variants independently is computationally prohibitive (Cai et al., 2020). This is particularly true for FL, where training these variants independently will cumulatively inflate the communication cost as well as the computation cost. Thus, it is imperative to develop techniques for training multiple models in a federated fashion cost efficiently without any accuracy loss—achieving asymptotic cost improvements relative to independently training them._ (Section 1, page 2, para 3)

---

> > > > > ### Comment · Reviewer_8Rfx · 2022-11-28
> > > > > **Response**
> > > > >
> > > > > Thank you for the response. With respect to the cost, could you clarify what the x-axis is in Figure 2?
> > > > >
> > > > > As for datasets, I view label-induced heterogeneity as synthetic heterogeneity, although it may still be useful. Furthermore, while BERT has been used in recent FL work (although notably not in the LEAF evaluations I pointed out), this does not preclude using TCNs there.

---

> > > > > > ### Author Response · Authors · 2022-11-28
> > > > > > **Response to Reviewer 8Rfx**
> > > > > >
> > > > > > Thanks!
> > > > > >
> > > > > > > could you clarify what the x-axis is in Figure 2?
> > > > > >
> > > > > > The x-axis represents FLOPs of subnetworks trained independently using FedAvg and in SuperFed. The goal of the experiment shown in Fig2 is not to show cost improvement but to demonstrate that SuperFed matches test accuracy/perplexity compared to independently trained networks for the entire chosen flop range.  The subnetworks are chosen such that they cover the entire FLOP range. The largest, smallest and middle subnetworks are included in Fig2. This captures the maximum and minimum achievable accuracy/perplexity.
> > > > > >
> > > > > > > I view label-induced heterogeneity as synthetic heterogeneity
> > > > > >
> > > > > > Label-induced heterogeneity is a standard for evaluating Federated Algorithms on non-iidness. Similar evaluation is done in [1,2,3,4,5,6,7,8,9,10]
> > > > > >
> > > > > >
> > > > > > > this does not preclude using TCNs there
> > > > > >
> > > > > > To re-iterate, we use PTB dataset as the official repo of TCN (https://github.com/locuslab/TCN) evaluates it on PTB dataset. We borrow the training recipes and hyper-parameters provided in the official code for performing evaluation in SuperFed. We further ask the reviewer to point us to any official code (and standard hyper-parameters) that trains TCN on LEAF datasets.
> > > > > >
> > > > > > ---
> > > > > > [1] https://proceedings.neurips.cc/paper/2020/hash/18df51b97ccd68128e994804f3eccc87-Abstract.html
> > > > > >
> > > > > > [2] https://proceedings.mlr.press/v97/yurochkin19a.html
> > > > > >
> > > > > > [3] https://openreview.net/forum?id=B7v4QMR6Z9w
> > > > > >
> > > > > > [4] https://openaccess.thecvf.com/content/CVPR2021/html/Li_Model-Contrastive_Federated_Learning_CVPR_2021_paper.html
> > > > > >
> > > > > > [5] https://proceedings.mlr.press/v162/kim22a.html
> > > > > >
> > > > > > [6] http://proceedings.mlr.press/v139/huang21c.html
> > > > > >
> > > > > > [7] https://openaccess.thecvf.com/content/CVPR2022/html/Gao_FedDC_Federated_Learning_With_Non-IID_Data_via_Local_Drift_Decoupling_CVPR_2022_paper.html
> > > > > >
> > > > > > [8] https://research.google/pubs/pub49350/
> > > > > >
> > > > > > [9] https://proceedings.neurips.cc/paper/2021/hash/2f2b265625d76a6704b08093c652fd79-Abstract.html
> > > > > >
> > > > > > [10] https://proceedings.neurips.cc/paper/2021/hash/46d0671dd4117ea366031f87f3aa0093-Abstract.html

---

### Official Review · Reviewer_nfyJ · 2022-10-29

**Confidence:** 3
**Correctness:** 3
**Technical Novelty And Significance:** 3
**Empirical Novelty And Significance:** 4
**Recommendation:** 6

**Clarity, Quality, Novelty And Reproducibility:**

The paper is well-motivated and the proposed solution has good intuitions. Comparisons to other baselines in some aligned setting can be considered.

The paper is clearly presented and very easy to read.

The paper studies a very novel problem setting 'training a global family of models'. However, maintaining many models in real-world applications can introduce a lot of complexities. Not sure whether it is practical in the real world.

**Strength And Weaknesses:**

Strength:
1) The proposed concept of 'training a global family of models' seems to be very well-motivated, novel and appealing.
2) The proposed method clearly outperforms the baseline, independent SOTA training of DNN subnetworks (without weight sharing) using FedAvg.

Weakness:
1) The concept of 'training a global family of models' is well-motivated by the need in on-device inference in real-world applications. However, by multiple model variants, I assume the variances of the performances in each devices can be much more significant, compared to the single model in previous FL approaches. However, only achieving some level of accuracy is not enough to release a real product (e.g, many human evaluations of the user experience are needed). With many different models in many devices, it will incur a lot of efforts to track and maintain these models. Thus, I am not sure whether the concept of 'training a global family of models' is practical in real-world applications.
2) Currently there is only one baseline evaluated in the paper. It is understandable that the paper proposes a very novel setting ('training a global family of models'), so the comparable baselines are limited. However, we may still align the setting and compare the proposed approach to existing FL approaches? For example, assume there are 10 models. For each client, we can train a suitable model by existing FL approaches on all clients with similar (or less) devices constraints. Although there are some benefits of 'training a global family of models', these experiments can still provide some insights about whether the proposed solution outperforms SOTA FL approaches (or still has a gap) in FL problems (e.g., non-iid).

**Summary Of The Paper:**

The paper proposes a weight shared training framework for FL that trains a family of model variants (DNN models), cost-efficiently in a federated fashion. The paper also proposes a heuristic that trains the upper and lower bounds in the model family by optimizing both bounds and load balancing their distribution over time. The paper further proposes a mechanism for subnetwork weight aggregation with variable overlaps mitigating intra-network interference. The proposed approach can take an important first step towards co-training model families in a federated fashion cost-efficiently, achieving an order of magnitude communication and computation cost reduction.

**Summary Of The Review:**

The paper studies a very novel problem setting 'training a global family of models'. However, maintaining many models in a real-world applications can introduce a lot of complexities. Not sure whether it is practical in the real world. Currently there is only one baseline evaluated in the paper. It is understandable that the paper proposes a very novel setting ('training a global family of models'), so the comparable baselines are limited. However, we may still align the setting and compare the proposed approach to existing FL approaches.

---

> ### Author Response · Authors · 2022-11-12
> **Response to Reviewer  nfyJ**
>
> We thank the reviewer for his valuable comments and appreciate that they find our work appealing. We provide clarification to the concerns raised by the reviewer -
> - **Practicality of Supernets in real world applications.** We emphasize that supernets may help in enhancing user experience which makes our proposed approach practical and relevant for real world applications. We agree with the reviewer that achieving some level of accuracy is not sufficient for user experience evaluation. In fact, there’s a class of applications that care about both accuracy and latency
>   - User experience not only depends on accuracy but also on satisfying dynamic deployment conditions such as making predictions within variable latency deadlines, consuming less battery power (to ensure the mobile device doesn’t abruptly terminate)or occupying fewer resources. For instance, in a google keyboard application, a fast typing user may have a better user experience with a faster model for next-word prediction. A slower typing user that may get a better user-experience from a slower but more accurate model.
> This ability to dynamically pick appropriate latency/accuracy tradeoffs at runtime is provided by the different subnetworks simultaneously trained as part of the weight shared supernet with SuperFed..
> - **Supernetworks are easy to maintain and deploy.** The reviewer raises a valid concern that a regular model zoo (collection of conventional model variants) may be tough to maintain and deploy in real world applications, however, we argue that this is not the case with the our chosen supernetwork mechanism of providing or encapsulating a model family -
>   - **O(1) Maintenance Cost at Deployment:** The maintenance of supernetwork at deployment is significantly simplified due to the following reasons :
>     - Tracking the largest and/or smallest subnetwork may be sufficient to track the entire range of subnetworks inside the supernetwork. This is because the weights are shared among all the subnetworks and the performance of lower and upper bound networks provides enough signal w.r.t tracking. Hence, in case of supernet, the tracking cost is O(1) i.e it doesn’t increase with the number of subnetworks being used (unlike regular model zoo).
>   - **Easy Deployment due to O(1) memory cost.** Supernetwork is easy to deploy as it occupies significantly less memory. It contains all subnetworks inside one large DNN and its memory doesn’t increase with the number of subnetworks used. This makes deployment of supernet easy especially when the resource-availability is dynamic (often the case in on-device inference).
> Different subnetworks can be activated from the supernetwork without any re-training. This allows clients to select subnetworks of their choice at any point of time without any additional memory overheads.
>
> - **Comparison of SOTA FL approaches.** We appreciate that the reviewer considers our FL setting novel and assert that advances made to FedAvg in FL to train a single model are complementary to SuperFed.
>   - For instance, many works like FedDyn, FedProx and Scaffold modify the client local training to improve FL on non-iid settings etc. FedProx adds a proximal term to the client local training loss. We believe such modifications can be done on top of our proposed SuperFed framework. Formulation of such weight-shared training algorithms can be pursued as an interesting future direction  (Sec 1.1).
>   - A fair apples to apples comparison will then be comparing modified weight-shared algorithms with their corresponding single model training algorithms.

---

### Official Review · Reviewer_EQW3 · 2022-11-03

**Confidence:** 2
**Correctness:** 3
**Technical Novelty And Significance:** 3
**Empirical Novelty And Significance:** 3
**Recommendation:** 6

**Clarity, Quality, Novelty And Reproducibility:**

Clarity - the paper is overall clear
Quality - the experiments are of strong quality on the presented datasets
Novelty - The paper effectively combines a number of existing ideas for a relevant proposed problem

**Strength And Weaknesses:**

Strengths

- The proposed problem (not directly studied in other works) is relevant
- the approach is relatively straightforward and elegant
- The results are convincing and well analyzed on the presented datasets

Weakness
- Although the datasets are standard for FL the authors primarily focus on variants of CIFAR-10 and convnets. It would be beneficial  to see the generality of the approach to other architectures (e.g. MLP's and ViT's) and more distinct datasets (e.g. text data).


**Summary Of The Paper:**

The paper proposes a weight shared training framework for FL that maintains a primary network at the server and distributes subnetworks to clients. The authors propose algorithms for efficient selection of the subnetworks and aggregation. This allows efficient federated learning and deployment of the models.


**Summary Of The Review:**

The paper provides an interesting and relevant problem setting and an effective algorithm for this setting. Validation on additional datasets and broad model classes would  increase the strength of the work.

---

> ### Author Response · Authors · 2022-11-12
> **Response to Reviewer EQW3**
>
> Thank you for your valuable suggestion. In the paper revision, we add a new experiment (Fig 2d) where we evaluate SuperFed (Maxnet) on the PennTreeBank (PTB) text dataset on a **new** architecture based on temporal convolution networks (https://arxiv.org/abs/1803.01271). Our experiments demonstrate that Maxnet achieves at-par test perplexity compared to the subnetworks individually trained using FedAvg on 20 clients with client participation ratio 0.4. We have thus been able to demonstrate the generalizability of the proposed approach not only to non-vision tasks, but also to other types of neural network architectures.

---

### Author Response · Authors · 2022-11-12
**Paper revision, key contributions and clarifications**

We thank the reviewers for their constructive comments. We’d like to start by restating our key contributions :
- SuperFed is a weight-shared training framework for FL that trains a family of model variants cost-efficiently for dynamic deployment conditions in on-device inference.
- Our proposed weight-shared training algorithm MaxNet innovates on subnetwork distribution and shared-parameter averaging.  MaxNet’s  subnetwork distribution optimizes the upper and lower bound of the model family with temporal load balancing. MaxNet’s shared-parameter averaging reduces training-time interference between co-trained model variants..
- MaxNet trains a family of ~5*10^18 subnets with **9.43x** lower computational and **10.94x** lower communication cost compared to training as few as 9 subnetworks separately with FedAvg.

We provide a paper revision in direct response to the reviewers’ valuable comments. Here are the key changes:
- **Generalizability on a Text Dataset.** We add a new experiment (Fig 2d) in the paper revision that evaluates SuperFed on PennTreebank dataset for next-word prediction task. The experiment is conducted on a supernet based on Temporal Convolution networks (https://arxiv.org/abs/1803.01271).
- **Code Release.** We include the reproducibility statement in the paper revision and provide anonymized source code (https://anonymous.4open.science/r/SuperFed/) that includes full instructions to perform weight-shared federated learning on the datasets and settings used in the paper.
- **Final Output of SuperFed.** At the end of training, the output of SuperFed is "k" global inference models, which is distinctly and fundamentally different from works in heterogeneous FL. HeteroFL eventually produces only a single global model to a given client. We clarify this important distinction in Sec 1.1 (revised). We thank anonReviewer4 for pointing out the typo and modify the text in Sec1.1 that removes the typo.

Finally, we’d like to factor out some key clarifications from individual responses that follow -
- **Generalizability on Text Dataset.** The new experiment demonstrates that Maxnet, the proposed weight-shared algorithm, indeed generalizes on a text dataset as well as a different NN architecture. It is able to match the test perplexity (defined as the exponential of the test loss) of independently trained subnetworks using FedAvg at a significantly reduced cost [anonReviewer1, anonReviewer3].
- **Differences b/w Heterogeneous FL and SuperFed.** We highlight three key differences between works in heterogeneous FL and SuperFed [anonReviewer4]
   - **Goal.** The goal of SuperFed is to train a global family of “k” models and provide them to each client for efficient on-device inference (post-training) . In sharp contrast, the goal of existing works in heterogeneous FL is to consider and manage resource constraints during training itself, not during inference
  - **Final Output.** At the end of training, SuperFed provides k global models for inference to each client whereas heterogeneous FL works provide a single global inference model to each client (a critical departure from our explicitly stated goal).
  - **Behavior in our setting.** In case of no resource constraints during training (the setting considered in SuperFed), heterogeneous FL works train only the largest network whereas SuperFed by design trains a family of models.
- **Practicality of Supernets in Real World Applications.** We make a case that supernetworks are practical in real world applications due to the following reasons  [anonReviewer2]  -
  - **Enhanced User Experience.** Supernets may enhance user experience as they satisfy dynamic deployment conditions at the time of on-device inference. For instance a fast-typing user with strict latency deadlines in google keyboard may get a better user experience in next word prediction due to supernets. This is because supernets allow activation of different subnetworks (without any retraining) dynamically depending on the deployment condition.
  - **Efficient Deployment due to O(1) Memory Cost.**  Supernets in deployment are memory efficient and take O(1) memory cost. This is because all subnetworks are contained inside and activated dynamically from one large DNN. This makes deployment of supernets significantly easier.
  - **O(1) Maintenance Cost at Deployment.** It is sufficient to track the largest and/or smallest subnetwork of the supernet in deployment. Hence, its maintenance cost doesn’t increase with the number of subnetworks used and is O(1).

---

### Public Comment · ~Adam1 · 2023-02-09
**interesting point**

Dear authors,
Hi, I've just ran into this paper and found the naming of the proposed method is exactly same with mine 😀 (except the letter 'p' should be uppercase in mine).
Here is my paper, introducing `SuPerFed` for enhanced personalized federated learning by inducing linear mode connectivity (LMC): https://arxiv.org/abs/2109.07628 (or https://dl.acm.org/doi/10.1145/3534678.3539254).

BTW, your findings and proposed method seems really interesting for improved FL for a practical use. I've enjoyed reading your work.
Just wanted to give you kudos and wish you luck!

Best,
Adam

---

> ### Author Response · Authors · 2023-02-09
> **thanks for appreciating and recognizing the value of our work**
>
> Hi Adam,
>
> Thanks for pointing out the name collision. Thankfully it's in the name only, as our works are quite distinct. We've been operating with this name for over a year now, so it was natural to keep it as is.
>
> Thanks a lot for your positive comments about this work. Yes, we also believe that this work will see significant adoption and recognition longer term, and we are happy that you've read it and can appreciate its value!

---

### Decision · Program_Chairs · 2023-01-20

**Decision:**

Reject

**Justification For Why Not Higher Score:**

While the authors propose a method that is considered novel by a subset of reviewers, all reviewers found the evaluation limited and incomplete. Accepting the paper in this form is not possible, unless a major revision is performed. We suggest a more thorough empirical evaluations, including applications both in NAS and FL scenarios.

**Justification For Why Not Lower Score:**

N/A

**Metareview: Summary, Strengths And Weaknesses:**

- Summary

The paper proposes a weight shared training framework for FL that trains a family of model variants (DNN models), cost-efficiently in a federated fashion. The paper also proposes a heuristic that trains the upper and lower bounds in the model family by optimizing both bounds and load balancing their distribution over time. The authors propose algorithms for efficient selection of the subnetworks and aggregation. This allows efficient federated learning and deployment of the models. The proposed approach can take an important first step towards co-training model families in a federated fashion cost-efficiently, achieving an order of magnitude communication and computation cost reduction.

- Strengths:

1. Reviewers found the idea of 'training a global family of models' well-motivated, and appealing.
2. Experiments show favorable performance for the selected (limited) tasks, supporting the algorithmic claims.
3. After the reviews, the authors have revised parts of the paper, by including additional experiments and code, along with explanations to reviewers' points.

- Weaknesses:

1. Reviewers still find the selection of the experiments provide partial answers to the efficacy and value of the algorithms. All reviewers appreciated the fact that the authors spent time to include experiments on the PTB dataset (which is a different application than the one included in the original version of the paper). Reviewers found this choice i) partially answers the concern of limited experiments (a reviewer mentions that in FL scenarios, more text-based datasets are available), ii) does not provide enough evidence that the methodology generalizes to other datasets + other tasks than the ones originally selected.

- Recommendation:

First of all, the authors should be stayed assured that there were discussions regarding the paper, and the area chair read carefully i) the paper, ii) the reviews, and iii) the discussions after the rebuttal. We acknowledge that the paper addresses concerns raised, but the reviewers have remaining concerns:

1. We recommend the authors include a more thorough experimental validation for their methodology: given that their method lies at the intersection of NAS and FL applications, including experiments that are common in both fields should be sufficient to answer any questions about the empirical validation of the method.
2. The paper lacks theoretical justification: while this is not necessary (and in some cases almost impossible), the experimental part should be complete, supporting further the point above. Experiments is what is only left for a reviewer to understand whether there is value in the methodology proposed;

We recommend the authors to consider these (common over all reviewers) concerns to revise their paper; we strongly support for a resubmission to a near-future ML conference.

---

> ### Author Response · Authors · 2023-01-22
> **can you enumerate explicitly what experiments you think are missing?**
>
> Thank you for your consideration and your service to the community. We have carefully read your justification for rejecting this paper. Please rest assured that we also take all reviewer comments very seriously. To follow up on your suggestion to strengthen the evaluation of this work, we would like to kindly ask for you to be very explicit about the kinds of experiments you think are lacking in this paper? Is there any chance the reviewers and/or the AC could enumerate, explicitly, the experiments you would like to see? We would very much like for this feedback to be _actionable_.
>
> * Is it not the case that we added all the experiments that have been requested by reviewers during the rebuttal period?
> * Is it not the case that we provided sufficient experimental evaluation to validate the **explicitly** stated hypothesis of this work?
> * Is it not the case that we demonstrated generalizability of this work to non-vision tasks, which extends beyond the initially stated hypothesis and scope?